# Visibility-Uncertainty-guided 3D Gaussian Inpainting via Scene Conceptional Learning

## ABSTRACT

3D Gaussian Splatting (3DGS) has emerged as a powerful and efficient 3D representation for novel view synthesis. This paper extends 3DGS capabilities to inpainting, where masked objects in a scene are replaced with new contents that blend seamlessly with the surroundings. Unlike 2D image inpainting, 3D Gaussian inpainting (3DGI) is challenging in effectively leveraging complementary visual and semantic cues from multiple input views, as occluded areas in one view may be visible in others. To address this, we propose a method that measures the visibility uncertainties of 3D points across different input views and uses them to guide 3DGI in utilizing complementary visual cues. We also employ uncertainties to learn a semantic concept of scene without the masked object and use a diffusion model to fill masked objects in input images based on the learned concept. Finally, we build a novel 3DGI framework, VISTA, by integrating VISibility-uncerTainty-guided 3DGI with scene conceptuAl learning. VISTA generates high-quality 3DGS models capable of synthesizing artifact-free and naturally inpainted novel views. Furthermore, our approach extends to handling dynamic distractors arising from temporal object changes, enhancing its versatility in diverse scene reconstruction scenarios. We demonstrate the superior performance of our method over state-of-the-art techniques using two challenging datasets: the SPIn-NeRF dataset, featuring 10 diverse static 3D inpainting scenes, and an underwater 3D inpainting dataset derived from UTB180, including fast-moving fish as inpainting targets.

## 1 INTRODUCTION

3D representation effectively models a scene and has the ability to synthesize new views of the scene (Barron et al., 2021; Mildenhall et al., 2021; Wang et al., 2021; Kerbl et al., 2023). 3D Gaussian splatting (3DGS) methods have been demonstrated as efficient and effective ways to represent the scene from a set of images taken from different viewpoints (Kerbl et al., 2023; Tang et al., 2023; Wu et al., 2024). Further, enabling editability of 3D scene representations is a cornerstone of technologies like augmented reality and virtual reality Tewari et al. (2022). 3D Gaussian inpainting task is one of the key editing techniques, aiming to replace specified objects with new contents that blend seamlessly with the surroundings. This capability allows us to: *(1) Remove objects from static scenes:* given multi-view images, we can create a 3D representation that generates novel views with specific objects removed and believably filled in (Figure 1 (Upper)). *(2) Clean up dynamic scenes:* for scenes with moving elements like fish in water (see Figure 1 (Bottom)), we can build a 3D representation excluding these transient objects, enabling clear, consistent novel view synthesis.

However, such an important task is non-trivial and the key challenge is how to leverage the complementary visual and semantic cues from multiple input views. Intuitively, for a synthesized view, the ideal approach is to replace the targeted erasure region with the occluded content, which naturally completes the inpainting. The key information for this process lies within the other view images, where the obscured areas may be visible from different angles. However, how to utilize multi-view information effectively is still an open question. State-of-the-art works first remove the targeted erasure region-related Gaussians and fill the regions via 2D image inpainting method (Ye et al., 2024; Wang et al., 2024), which, however, neglects the complementary cues from other views. The latest work (Liu et al., 2024) leverages depth maps of different views to involve the cross-view complementary cues implicitly. However, depth maps cannot fully represent complementary cues, such

Figure 1: Two examples demonstrating two state-of-the-art methods, namely InFusion (Liu et al., 2024) and GaussianGroup (Ye et al., 2024), alongside our proposed method for 3D Gaussian inpainting to fill masked static and dynamic objects, respectively. The red boxes highlight the advantages of our method and are enlarged on the right side of each image for better visibility. The white boxes and arrows indicate complementary visual cues between two different viewpoints

as the texture pattern from adjacent perspectives, and the depth project can hardly get high-quality depth maps when moving objects across different views. As the two cases shown in Figure 1, InFusion synthesizes new views with obvious artifacts.

In this work, we propose VISibility-uncerTainty-guided 3DGI via scene conceptuAl Learning (VISTA), a novel framework for 3D Gaussian inpainting that leverages complementary visual and semantic cues. Our approach begins by measuring the visibility of 3D points across different views to generate visibility uncertainty maps for each input image. These maps indicate which pixels are most valuable for the inpainting task, based on the principle that pixels visible and consistent from multiple views contribute more significantly. We then integrate these visibility uncertainty maps into the 3D Gaussian splatting (3DGS) process. This enables the resulting Gaussian model to synthesize new views where masked regions are seamlessly filled with visual information from complementary perspectives. To address scenarios where large masked regions lack complementary visual cues from other views, we propose learning the concept of the scene without the masked objects. This conceptual learning is guided by the prior inpainting mask and the visibility uncertainty maps derived from the input multi-view images. The learned concept is then utilized to refine the input images, effectively filling the masked objects through a pre-trained Diffusion model. Furthermore, we implement an iterative process alternating between visibility-uncertainty-guided 3DGI and scene conceptual learning, progressively refining the 3D representation. As illustrated in Figure 1 (Upper), our method successfully reconstructs high-quality 3D representations of static scenes, naturally filling masked object regions with contextually appropriate content. Additionally, VISTA demonstrates its versatility by effectively removing distractors in dynamic scenes (see Figure 1 (Bottom) for examples).

We demonstrate the superior performance of our method over state-of-the-art techniques using two challenging datasets: the SPIn-NeRF dataset, featuring 10 diverse static 3D in-painting scenes, and an underwater 3D inpainting dataset derived from UTB180, which includes fast-moving fish as inpainting targets. In summary, the contributions of our work are as follows:

1. We propose VISibility-uncerTAinty-guided 3D Gaussian inpainting (VISTA-GI) that explicitly leverages multi-view information through visibility uncertainty, achieving 3D Gaussian inpainting for more coherent and accurate scene completions.

2. We propose VISibility-uncerTAinty-guided scene conceptual learning (VISTA-CL) and leverage it for diffusion-based inpainting. VISTA-CL fills masked regions in input images using learned scene concepts, addressing the inpainting task at its core. This approach enhances the fundamental understanding of the scene, leading to more accurate and contextually appropriate inpainting results.

3. We introduce VISTA (VISibility-uncerTainty-guided 3D gaussian inpainTing via scene conceptuAl learning), a novel framework that iteratively combines VISTA-GI and VISTA-CL. This approach simultaneously leverages complementary visual and semantic cues, enhancing 3D Gaussian inpainting with geometric and conceptual information.

4. We extend VISTA to handle dynamic distractor removal in 3D Gaussian splatting, significantly improving its performance on scenes with temporal variations and outperforming state-of-the-art methods.

## 2 RELATED WORK

### 2.1 NeRF AND 3D GAUSSIAN SPLATTING

Reconstructing 3D scenes from 2D images for novel view synthesis remains a fundamental challenge in computer vision and graphics (Lombardi et al., 2019; Kutulakos & Seitz, 2000). Recent advancements are driven by two contrasting paradigms: NeRF's implicit neural representations (Mildenhall et al., 2021) and 3DGS's explicit Gaussian modeling (Kerbl et al., 2023).

Neural Radiance Fields (NeRF) reconstruct scenes by learning continuous volumetric radiance fields through multi-view image optimization, achieving photorealistic view synthesis (Barron et al., 2021). However, its computational intensity in both training and rendering (Barron et al., 2022; 2023) limits practical deployment. In contrast, 3D Gaussian Splatting (3DGS) employs explicit geometric primitives - anisotropic Gaussian blobs with position, opacity, covariance, and color attributes (Lu et al., 2024). This approach achieves real-time rendering of high-fidelity novel views through direct spatial optimization of discrete 3D elements.

### 2.2 2D INPAINTING AND 3D INPAINTING

Image inpainting, a fundamental task in image generation, has evolved from traditional patch-based approaches (Ružić & Pižurica, 2014) to GAN-driven methods (Goodfellow et al., 2014; Yu et al., 2018), which handle regular regions but struggle with complex occlusions. Diffusion models now dominate this field (Ho et al., 2020; Sohl-Dickstein et al., 2015; Song et al., 2020), demonstrating superior capability in generating semantically coherent content for large missing areas (Lugmayr et al., 2022; Suvorov et al., 2022; Li et al., 2022).

Extending inpainting to 3D neural representations presents unique challenges. While NeRF-based approaches (Liu et al., 2022; Mirzaei et al., 2023; Weder et al., 2023) achieve partial success with static volumetric representations, their effectiveness remains constrained by inherent architectural limitations. Explicit 3DGS inpainting methods like Gaussian Grouping (Ye et al., 2024), InFusion (Liu et al., 2024), and GaussianEditor (Wang et al., 2024) primarily address static scene editing, overlooking dynamic interference during scene capture. Recent works attempt new directions: GScream (Wang et al., 2025) integrates monocular depth estimation with cross-attention for object removal, while SpotLessSplats (Sabour et al., 2024) addresses dynamic distractors through mask propagation - though both struggle with reconstructing heavily occluded regions.

## 3 PRELIMINARIES: 3D GAUSSIAN SPLATTING AND INPAINTING

### 3.1 3D GAUSSIAN SPLATTING

Given a set of images $\mathcal{I} = \{\mathbf{I}_i\}_{i=1}^N$ captured from various viewpoints and timestamps, 3D Gaussian splatting (3DGS) aims to learn a collection of anisotropic Gaussian splats $\mathcal{G} = \{\mathbf{g}_j\}_{j=1}^M$ from these multi-view images. Each splat $\mathbf{g}_j$ is characterized by a Gaussian function with mean $\mu_j$, a positive semi-definite covariance matrix $\sum_j$, an opacity $\alpha_j$, and view-dependent color coefficients $\mathbf{c}_j$. Once the parameters of the 3D Gaussian splats $\mathcal{G}$ are determined, novel view synthesis can be achieved through alpha-blending: $\hat{\mathbf{I}}^p = \text{Render}(\mathcal{G}, \mathbf{p})$. We can use $\mathcal{I}$ to supervise the optimization of $\mathcal{G}$

$$\arg\min_{\mathcal{G}} \lambda_1 \sum_{i=1}^N \|(\mathbf{I}_i - \hat{\mathbf{I}}^{p_i})\|_1 + \lambda_2 \sum_{i=1}^N \text{D-SSIM}(\mathbf{I}_i, \hat{\mathbf{I}}^{p_i}), \tag{1}$$

where $\mathbf{p}_i$ denotes the camera perspective of the image $\mathbf{I}_i$, $\hat{\mathbf{I}}^{p_i} = \text{Render}(\mathcal{G}, \mathbf{p}_i)$, and $\lambda_1 + \lambda_2 = 1$. For novel view synthesis, given a camera perspective $\mathbf{p}$, the process involves the following steps: projecting each 3D Gaussian onto a 2D image plane, sorting the Gaussians by depth along the view direction, and blending the Gaussians from front to back for each pixel. A key advantage of 3DGS

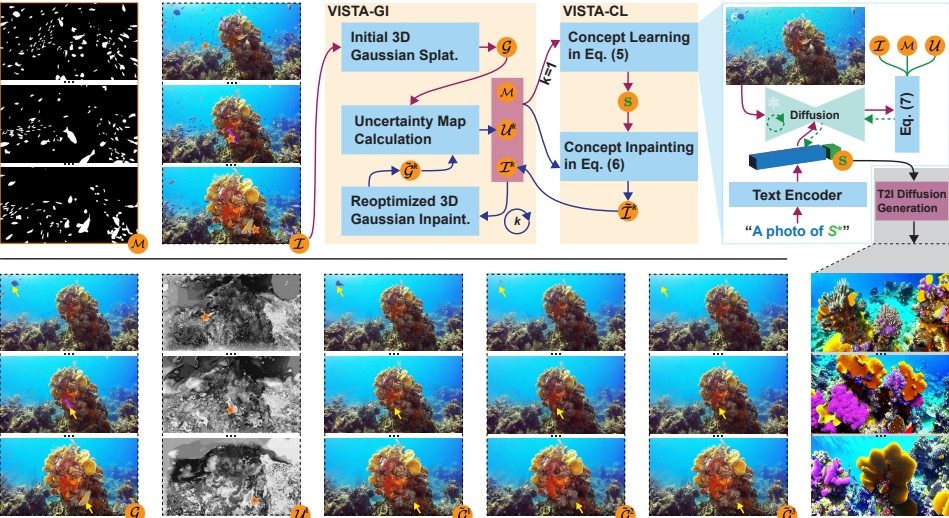

Figure 2: The framework of VISTA comprises two modules: VISTA-GI (described in Section 4.1) and VISTA-CL (detailed in Section 4.2). Results from three views are displayed for key variables in the framework. Note that $\mathcal{G}$, $\tilde{\mathcal{G}}^1$, $\tilde{\mathcal{G}}^2$, and $\tilde{\mathcal{G}}^3$ are 3DGS representations, and the displayed examples are rendered from these representations. The last column shows generated images derived from the learned scene concept. In the uncertainty map, we use ☆ to highlight areas of high uncertainty, which denote points (e.g., dynamic fishes) visible from only a few views. Yellow arrows demonstrate the progressive improvement in inpainting quality achieved by our method.

(Kerbl et al., 2023) is its ability to synthesize a new view in a single pass, whereas NeRF requires pixel-by-pixel rendering. This efficiency makes 3DGS particularly well-suited for time-sensitive 3D representation applications, offering a significant performance boost over NeRF.

## 3.2 3D GAUSSIAN INPAINTING

Given a set of captured images $\mathcal{I} = \{\mathbf{I}_i\}_{i=1}^N$ and corresponding binary mask maps $\mathcal{M} = \{\mathbf{M}_i\}_{i=1}^N$ delineating objects for removal (See Figure 1), 3D Gaussian Inpainting (3DGI) constructs a new 3D Gaussian splatting (3DGS) representation. This representation eliminates specified objects and replaces them with content that integrates with the environment. The resulting 3DGS representation can synthesize arbitrary views where the specified objects are imperceptibly absent, maintaining visual coherence across viewpoints while effectively 'erasing' targeted objects. We can use the segment anything model (SAM) (Kirillov et al., 2023) with few manual annotations to generate mask maps, aligning with methods like (Ye et al., 2024) for precise object delineation.

**SOTA methods and limitations.** An intuitive approach to 3D Gaussian Inpainting (3DGI) involves deriving a 3D mask for the specified objects based on the provided 2D masks. The process of new view synthesis then follows a two-step procedure: first, generating the specified view and its corresponding mask, and then applying existing 2D image inpainting techniques to achieve the desired 3DGI effect. This methodology has been adopted in recent works by Wang et al. (2024) and Ye et al. (2024). However, this approach does not leverage the complementary information available across multiple viewpoints during the inpainting process. A key example is the failure to utilize information from regions that may be occluded in one view but visible in another. Consequently, this method struggles to maintain consistency with the surrounding environment, particularly when dealing with large masked regions. This limitation underscores the need for more sophisticated techniques to effectively integrate and synthesize information from multiple perspectives to achieve more coherent and realistic 3D inpainting results. Beyond this solution, the latest work Liu et al. (2024) utilizes the cross-view complementary cues through depth perception. It formulates the 3D Gaussian inpainting as two tasks, *i.e.*, 2D image inpainting and depth inpainting, and the complementary cues in multiple views are implicitly utilized via depth projection. However, depth maps cannot fully represent complementary cues, such as the texture pattern from adjacent perspectives, and the depth project can hardly get high-quality depth maps when moving objects across different views. As case 2 shown in Figure 1, InFusion synthesizes new views with obvious artifacts.

## 4 METHODOLOGY

This section details the proposed framework called VISibility-uncerTainty-guided 3D Gaussian inpainting via scene concepTional learning (VISTA). The core principle is to identify the visibility of 3D points across different views and utilize this information to guide the use of complementary visual and semantic cues for 3D Gaussian inpainting.

To elucidate this concept, we introduce the visibility-uncertainty-guided 3D Gaussian inpainting (VISTA-GI) in Section 4.1, where we define the visibility uncertainty of 3D points and employ it to guide the use of complementary visual cues for 3DGI. In Section 4.2, we propose leveraging the visibility uncertainty to learn the semantic concept of the scene without specified objects. We then perform concept-driven Diffusion inpainting to process the input images, harnessing complementary semantic cues. To fully utilize complementary visual and semantic cues, we propose in Section 4.3 an iterative combination of VISTA-GI and VISTA-CL. Finally, in Section 4.4, we extend our VISTA framework to address the challenge of dynamic distractors in captured images. This extension excludes transient objects, resulting in clearer and more consistent novel view synthesis.

### 4.1 VISTA-GI: VISIBILITY-UNCERTAINTY-GUIDED 3D GAUSSIAN INPAINTING

**Initial 3D Gaussian Splatting.** Given the input images $\mathcal{I} = \{\mathbf{I}_i\}_{i=1}^N$, we employ the original 3DGS method in Section 3.1 and Equation (1) to construct a 3D representation $\mathcal{G}$. This representation can then be utilized to render novel views. However, as illustrated in Figure 2, this initial representation fails to exclude dynamic objects (such as fish) and exhibits noticeable artifacts, including blurring.

**Visibility uncertainty of 3D Points.** We define a set of adjacent camera views (indicating the top-K nearest cameras, determined by the distance of camera centers), denoted as $\mathcal{P} = \{\mathbf{p}_v\}_{v=1}^V$, where $V$ is the number of adjacent views. For a 3D point $\mathbf{X}$ in the scene, we can project it to different camera perspectives in $\mathcal{P}$ via the built 3DGS $\mathcal{G}$ and get their colors under $V$ views, *i.e.*, $\{\mathbf{x}_v\}_{v=1}^V$. Then, we calculate the variations of the colors of the point under different views

$$u_{\mathbf{x}} = \text{var}(\{\mathbf{x}_v\}_{v=1}^V), \tag{2}$$

where $\text{var}(\cdot)$ is the variation function. We denote the result $u_{\mathbf{x}}$ as the *visibility uncertainty* of the 3D point $\mathbf{X}$. Intuitively, $u_{\mathbf{x}}$ represents the visibility and consistency of the point across the $V$ views. For example, if the point $\mathbf{X}$ can be seen at all views, the colors under different views are consistent and $u_{\mathbf{x}}$ is small. If the point can be only seen by a few views or its color deviates between different views, the visibility uncertainty tends to be significantly high.

**Reoptimized 3D Gaussian inpainting.** With the 3D point's visibility uncertainty, we aim to calculate the visibility uncertainty map of the input image and measure the visibility of each pixel at other views. Specifically, for an image $\mathbf{I}_i$ in $\mathcal{I}$, we first calculate its depth map $\mathbf{D}_i$ based on the $\mathcal{G}$. Then, we project each pixel of $\mathbf{I}_i$ to a 3D point and calculate its visibility uncertainty via Equation (2) under $V$ adjacent views. Then, we obtain a pixel-wise visibility uncertainty map, which is normalized by dividing each pixel's uncertainty value by the standard deviation computed across all uncertainty values. The resulting normalized map is denoted as $\mathbf{U}_i$. For the $N$ input images, we have $N$ visibility uncertainty maps $\mathcal{U} = \{\mathbf{U}_i\}_{i=1}^N$. Then, we use them to update the original mask maps $\mathcal{M}$ and uncertainty maps $\mathcal{U}$ by

$$\mathbf{M}_i' = \mathbf{U}_i \odot (1 - \mathbf{M}_i) + \vartheta \cdot \mathbf{M}_i, \tag{3}$$

where the first term weights the unmasked regions via the visibility uncertainty map: the points other views cannot see should be assigned low weights during optimization. The $\vartheta$ controls the constraint degrees of the original masks. Then, we obtain the finer mask maps $\{\mathbf{M}_i'\}_{i=1}^N$ and re-optimize the 3D representation by adding the guidance of mask maps to the objective function in Equation (1):

$$\arg\min_{\mathcal{G}} \lambda_1 \sum_{i=1}^N \|(1 - \mathbf{M}_i') \odot (\mathbf{I}_i - \hat{\mathbf{I}}^{p_i})\|_1 + \lambda_2 \sum_{i=1}^N \text{D-SSIM}(\mathbf{I}_i, \hat{\mathbf{I}}^{p_i}, 1 - \mathbf{M}_i'), \tag{4}$$

where we have $\hat{\mathbf{I}}^{p_i} = \text{Render}(\mathcal{G}, \mathbf{p}_i)$ and $\lambda_1 + \lambda_2 = 1$. Intuitively, the objective function is to ignore the mask and high-uncertainty regions during the optimization. As a result, we get an updated counterpart $\tilde{\mathcal{G}}$. Similar strategies have been also adopted in recent works (Sabour et al., 2024; 2023).

Intuitively, with the visibility uncertainty maps, we can exclude the pixels that other views cannot see to build the 3D representation, which explicitly leverages the complementary visual cues. As the $\mathcal{U}$ shown in Figure 2 (Bottom) , the pixels with high uncertainty denote the corresponding points (e.g., dynamic fishes) visible from only a few views. This is reasonable since the dynamic fishes are at different locations across different views. We also display the updated 3D representation $\tilde{\mathcal{G}}^1$, showing that the dynamic objects and some artifacts are removed.

## 4.2 VISTA-CL: VISIBILITY-UNCERTAINTY-GUIDED SCENE CONCEPTUAL LEARNING

VISTA-GI can reconstruct masked objects when complementary visual information is available from alternative viewpoints. However, for masked regions lacking such cues, we need a more sophisticated approach to comprehend the scene holistically and generate plausible new content to fill these gaps. To achieve this, we propose to learn a conceptual representation $\mathbf{s}$ of the scene through textual inversion (Gal et al., 2022; Zhu et al., 2024), which can be formulated as

$$\mathbf{s} = \text{ConceptLearn}(\mathcal{I}, \mathcal{U}, \mathcal{M}), \tag{5}$$

The learned concept $\mathbf{s}$ is a token and encapsulates the scene's essence without the masked objects. We then leverage $\mathbf{s}$ to process the input images, eliminating the masked objects

$$\tilde{\mathbf{I}}_i = \text{ConceptInpaint}(\mathbf{s}, \mathbf{I}_i, \mathcal{U}, \mathcal{M}), \forall \mathbf{I}_i \in \mathcal{I}, \tag{6}$$

**Scene conceptual learning.** We formulate the scene conceptual learning, *i.e.*, as the personalization text-to-image problem (Ruiz et al., 2023) based on textual inversion (Gal et al., 2022), and we add the guidance of the visibility uncertainty maps in Section 3.2. Specifically, we have a pre-trained text-2-image diffusion model containing an image autoencoder with $\phi$ and $\phi^{-1}$ as encoder and decoder, a text encoder $\varphi$, and a conditional diffusion model $\epsilon_\theta$ at latent space. Then, we learn the scene concept $\mathbf{s}$ by optimizing the following objective function

$$\mathbf{s} = \underset{\mathbf{s}^*}{\arg\min} \mathbb{E}_{\mathbf{I}_i \in \mathcal{I}, \mathbf{z} = \phi(\mathbf{I}), \mathbf{y}, \epsilon \in \mathcal{N}(0,1), t}(\|(1 - \mathbf{M}_i') \odot (\epsilon_\theta(\mathbf{z}_t, t, \Upsilon(\varphi(\mathbf{y}), \mathbf{s}^*)) - \epsilon)\|_2^2), \tag{7}$$

where $\mathbf{y}$ is a fixed text (*i.e.*, 'a photo of $S^*$') and the function $\Upsilon(\Gamma(\mathbf{y}), \mathbf{s}^*)$ is to replace the token of '$S^*$' within $\Gamma(\mathbf{y})$ with $\mathbf{s}^*$. The tensor $\mathbf{M}_i'$ is calculated via Equation (3) based on the visibility uncertainty map and the given mask map. Intuitively, we use the Equation (7) to force the learned concept to mainly contain the unmasked scene regions. To validate the learned concept, we can feed 'a photo of $S^*$' to the T2I diffusion model to generate images about the learned concept. As shown in Figure 2, the images in the lower right are created directly by the T2I diffusion model and illustrate a concept similar to the original scene without any dynamic objects.

**Scene conceptual-guided inpainting.** We use the learned concept $\mathbf{s}$ to inpaint all input images through the pre-trained T2I diffusion model. Given one image $\mathbf{I}$ from $\mathcal{I}$, we can extract its latent code by $\mathbf{z} = \phi(\mathbf{I})$. Then, we perform the forward diffusion process by iteratively adding Gaussian noise to the $\mathbf{z}$ over $T$ timesteps, obtaining a sequence of noisy latent codes, *i.e.*, $\mathbf{z}_0$, $\mathbf{z}_1$, ..., $\mathbf{z}_T$, where $\mathbf{z}_0 = \mathbf{z}$. At the $t$th step, the latent is obtained by

$$q(\mathbf{z}_t|\mathbf{z}_0) = \sqrt{\overline{\alpha}_t}\mathbf{z}_0 + \sqrt{1 - \overline{\alpha}_t}\epsilon_t, \ \epsilon_t \sim \mathcal{N}(0, \mathbb{I}), \tag{8}$$

where $\overline{\alpha}_t = \prod_{\tau=1}^t (1-\beta_\tau)$. $\mathcal{N}(0, \mathbb{I})$ represents the standard Gaussian distribution. As we set the time step as $T$, the complete forward process can be expressed as $\mathbf{z}_T \sim q(\mathbf{z}_{1:T}|\mathbf{z}_0) = \prod_{t=1}^T q(\mathbf{z}_t|\mathbf{z}_{t-1})$.

At the reverse denoising process, we follow the strategy of RePaint (Lugmayr et al., 2022) but embed the guidance of visibility uncertainty maps and the learned concept $\mathbf{s}$. Intuitively, at the time step $t > 1$ during denoising, we only denoise the masked regions conditioned on the scene concept $\mathbf{s}$ while maintaining the unmasked regions with the same content in Equation (8), that is, we have

$$\tilde{\mathbf{z}}_{t-1} = (1 - \mathbf{m}') \odot \mathbf{z}_{t-1} + \mathbf{m}' \odot \hat{\mathbf{z}}_{t-1}, \tag{9}$$

where $\mathbf{z}_{t-1} \sim q(\mathbf{z}_t|\mathbf{z}_0)$ and $\mathbf{m}'$ is the downsampled $\mathbf{M}' \in \{\mathbf{M}_i'\}_{i=1}^N$ calculated by Equation (3) and has the exact resolution as the latent code $\mathbf{z}_{t-1}$. $\hat{\mathbf{z}}_{t-1}$ is denoised from the $\tilde{\mathbf{z}}_t$ with the guidance of the learned concept $\mathbf{s}$, that is,

$$\hat{\mathbf{z}}_{t-1} = \frac{1}{\sqrt{\alpha_t}}(\tilde{\mathbf{z}}_t - \frac{\beta_t}{\sqrt{1 - \overline{\alpha}_t}}\epsilon_\theta(\tilde{\mathbf{z}}_t, t, \mathbf{s})) + \sigma_t\xi, \text{s.t.}, \xi \sim \mathcal{N}(0, \mathbb{I}), \tag{10}$$

If $t = 1$, $\tilde{\mathbf{z}}_0 = (1 - \mathbf{m}') \odot \mathbf{z} + \mathbf{m}' \cdot \hat{\mathbf{z}}_0$. Then, we can get the inpainted image via decoder $\tilde{\mathbf{I}} = \phi^{-1}(\tilde{\mathbf{z}}_0)$. We can use the above ConceptInpaint to process each image within $\mathcal{I}$ and get a new image set $\tilde{\mathcal{I}}$.

### 4.3 VISTA: COMBINING VISTA-GI AND VISTA-CL

Given the input images $\mathcal{I}$ and their corresponding mask maps $\mathcal{M}$, VISTA-GI generates the uncertainty maps $\mathcal{U}$ as the visual cues and refines the 3DGS representation $\tilde{\mathcal{G}}$. VISTA-CL takes $\mathcal{I}$, $\mathcal{U}$, and $\mathcal{M}$ as inputs and produces processed input images $\tilde{\mathcal{I}}$ as the semantic cues. Intuitively, we can combine the raw images $\mathcal{I}$ and $\tilde{\mathcal{I}}$, feed them back into VISTA-GI, where $\tilde{\mathcal{I}}$ serve as better views. This allows for iterative process between VISTA-GI and VISTA-CL. We denote the $k$-th iteration's 3D representation from VISTA-GI as $\tilde{\mathcal{G}}^k$ and the processed images from VISTA-CL as $\tilde{\mathcal{I}}^k$.

In practice, three iterations are typically sufficient to achieve smooth convergence of the training metrics. The hyperparameter $\vartheta$ is initialized by 0 and increases by 0.1 with each iteration. We show an example in Figure 2. The synthetic views $\tilde{\mathcal{G}}^1$, $\tilde{\mathcal{G}}^2$, and $\tilde{\mathcal{G}}^3$ gradually contain fewer distractors, and the results of the final iteration $\tilde{\mathcal{G}}^3$ demonstrate clean and clear views, which means better 3D inpainting under the guidance of the visual and semantic cues.

### 4.4 VISTA FOR DYNAMIC DISTRACTOR REMOVAL

VISTA could be easily extended to remove dynamic distractors across multi-view images $\mathcal{I}$ by identifying the dynamic regions in $\mathcal{I}$ and obtaining the mask maps $\mathcal{M}$. In our implementation, we use the tracking method and MASA (Li et al., 2024) to automatically get the mask maps for dynamic objects in the scene. MASA is an open-vocabulary video detection and segmentation model introducing coarse pixel-level information to our method. This plays a similar role as DEVA (Cheng et al., 2023) used in Gaussian Grouping (Ye et al., 2024). However, the masks used in Gaussian Grouping are limited to static objects, while we mask static and dynamic objects that need to be inpainted. For dynamic objects, the uncertainty map can complement the coarse mask that excludes those dynamic distractors from the reconstruction. As shown in Figure 2, the synthetic view $\mathcal{G}$ obtained without masks fairly removes those fish moving greatly but ignores those objects without significant movement. The semantic information in the coarse masks $\mathcal{M}$ identifies these distractors, which the uncertainty map $\mathcal{U}$ cannot detect, and then these distractors can be eliminated by VISTA-CL. As a result, VISTA can remove both static and dynamic distractors in the scene by combining these two mask maps in Equation (3).

## 5 EXPERIMENTS

### 5.1 DATASETS AND METRICS

To evaluate our method, we conduct experiments on the SPIn-NeRF Dataset and the Underwater 3D Inpainting Dataset for scene repair in different scenarios.

**Underwater 3D inpainting dataset.** This dataset is derived from the underwater object tracking dataset UTB180 (Alawode et al., 2022), from which we selected multiple videos for resampling, ultimately forming 10 underwater scene datasets. We resample the video at a certain FPS to fulfill the motion requirements of the initial reconstruction. Each scene con-

| Method | UCIQE ↑ | URanker ↑ | CLIP Score ↑ | Rank-1 ↑ |
|---|---|---|---|---|
| SPIn-NeRF | 0.49 | 1.59 | 0.70 | 15.89 % |
| InFusion | 0.50 | 1.52 | 0.71 | 3.39 % |
| SpotLess | 0.50 | 1.59 | 0.70 | 9.63 % |
| Ours | **0.51** | **1.64** | **0.72** | **71.09 %** |

Table 1: Results of dynamic inpainting on the Underwater 3D Inpainting Dataset.

tains dozens of images from various viewpoints, and the initial Structure from Motion point cloud and camera intrinsics are obtained via COLMAP (Schonberger & Frahm, 2016). Each viewpoint image undergoes object detection using the open-source method MASA (Li et al., 2024) to obtain rough object masks.

**SPIn-NeRF dataset.** The SPIn-NeRF dataset was proposed in Mirzaei et al. (2023). It contains 10 general 3D inpainting scenes, divided into 3 indoor and 7 outdoor scenes. Each scene includes 100 images from various viewpoints, along with corresponding masks. In these datasets, the ratio of the training set to the

| Method | LPIPS ↓ | Fid ↓ | PSNR ↑ | SSIM ↑ |
|---|---|---|---|---|
| Masked Gaussians | 0.594 | 278.32 | 10.77 | 0.29 |
| SPIn-NeRF | 0.465 | 156.64 | 15.80 | 0.46 |
| Gaussians Grouping | 0.454 | 123.48 | 14.86 | 0.27 |
| GScream | 0.422 | 114.41 | 15.98 | **0.59** |
| InFusion | 0.567 | 118.26 | 15.59 | 0.53 |
| Ours | **0.418** | **113.58** | **16.48** | **0.59** |

Table 2: Static inpainting for SPIn-NeRF Dataset.

testing set is 6 to 4. We compare our method with other approaches using the provided camera intrinsics and an initialized SfM point cloud.

**Metrics.** Following SPIn-NeRF, we evaluate results using two quantitative protocols: (1) for static scenes with ground truth, we adopt PSNR, SSIM, LPIPS, and FID for reference-based image quality assessment (IQA); (2) for dynamic scenes without ground truth, we employ UCIQE (Yang & Sowmya, 2015), URanker (Guo et al., 2023), and CLIP Score (Hessel et al., 2021) for underwater non-reference IQA. Consistent with SPIn-NeRF (Mirzaei et al., 2023) and RefFusion (Mirzaei et al., 2024), LPIPS and FID are computed within the masked region defined by its bounding box. UCIQE combines chroma, saturation, and contrast to quantify color casts, blurriness, and low contrast; URanker is a transformer-based underwater image quality metric; CLIP Score measures image–text alignment, using the caption *"An underwater scene without fish"* to assess fish removal. To address the insensitivity of non-reference metrics, we conducted a user study on 48 images randomly sampled from six scenes. For each case, participants compared renderings from all methods and selected the one with the highest visual quality. The *Rank-1* column reports the proportion of cases in which a method was chosen as the top-performing approach.

## 5.2 EXPERIMENTAL RESULTS

We compare our method with several state-of-the-art open-source 3D inpainting methods, such as Infusion (Liu et al., 2024), SPIn-NeRF (Mirzaei et al., 2023), Gaussian Grouping (Ye et al., 2024), and SpotLessSplats (Sabour et al., 2024). SpotLessSplats is only designed for scenarios with dynamic distractors, while others are the latest static inpainting methods. Infusion (Liu et al., 2024) and GScream (Wang et al., 2025) are retrained using their publicized code. Further ablation studies and detailed discussions are presented in Appendix A.

**Results on the underwater 3D inpainting dataset.** Our method demonstrates superior underwater inpainting performance in Figure 3, achieving artifact-free 3D consistency across dynamic objects where baselines fail: SpotLessSplats incompletely removes moving fish Gaussians; Infusion induces viewpoint distortions despite localized clarity; SPIn-NeRF maintains 3D coherence but exhibits artifacts. Quantitative validation in Table 1 reveals our approach surpasses competitors in UCIQE and URanker metrics through uncertainty-guided

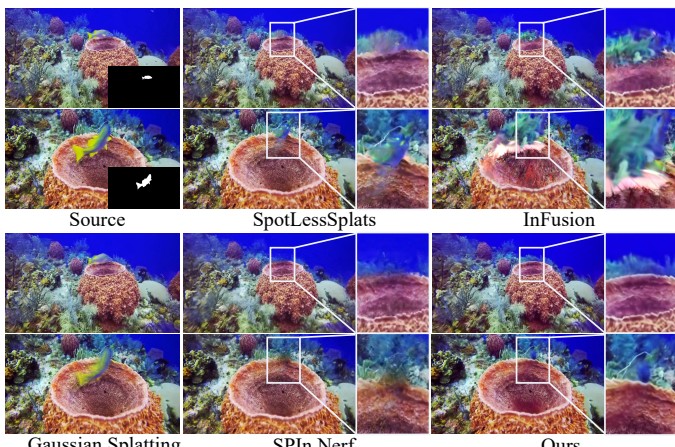

Figure 3: Visualization for dynamic inpainting.

reconstruction, while superior CLIP Scores confirm effective target object removal through optimized semantic-geometric integration. The 71.09% rank-1 preference—significantly higher than baselines—aligns with visual evidence (e.g., Figure 3) and underscores our method's effectiveness in perceptually critical areas. While standard non-reference metrics indicate modest gains, human evaluation confirms that our method delivers the best reconstruction in structural coherence and detail recovery.

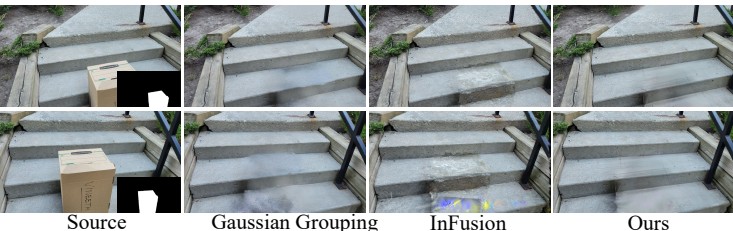

Figure 4: Visualized examples of static inpainting on SPIn-NeRF.

**Results on SPIn-NeRF dataset.** Figure 4 depicts an example scene from the SPIn-NeRF Dataset masking a stationary box that requires inpainting. The results of Gaussian Grouping are fairly realistic at the 2D image level, but there are significant inconsistencies between perspectives, such as distortion at the edges of stairs. The results of InFusion appear more realistic from a certain perspective. Still, its approach of optimizing one single view compromises the performance of other perspectives, leading to unpredictable artifacts in those views. As shown in Table 2, our method benefits from an iterative progressive optimization approach, ensuring consistency across perspectives through multiple inpainting and reconstruction, resulting in more stable outcomes.

**Ablation study on VISTA-GI and VISTA-CL.** We conducted the ablation study on the underwater 3D inpainting dataset by removing the VISTA-GI and VISTA-CL from our pipeline respectively. The specific results are shown in Table 3. Our experiments reveal two main findings: 1) Using only a 2D generative model with-

| Method | UCIQE ↑ | URanker ↑ | CLIP Score ↑ |
|---|---|---|---|
| Ours w/o VISTA-GI | 0.48 | 1.52 | 0.70 |
| Ours w/o VISTA-CL | 0.50 | 1.59 | 0.69 |
| Ours | **0.51** | **1.64** | **0.72** |

Table 3: Ablation study of VISTA-GI and VISTA-CL on the Underwater 3D Inpainting Dataset.

out VISTA-GI results in poor metrics, confirming that VISTA-GI's uncertainty guidance reduces multi-view inconsistencies in 3D reconstruction, leading to better outputs. 2) Omitting VISTA-CL maintains image quality similar to methods like SplotLess and SPIn-NeRF, but significantly lowers CLIP-Score metrics. This shows that without conceptual constraints, the inpainting process is visually plausible but semantically inconsistent with the scene context.

**Impact of noise reduction ratios in diffusion.** We analyze noise reduction strategies during diffusion model inference by implementing fixed-ratio iterative noise reduction from 1.0, evaluating four ratios $\{0.1, 0.2, 0.3, 0.4\}$ on reconstruction quality. As shown in Figure 5 (a), all ratios improve PSNR iteratively, but the 0.2 ratio achieves optimal convergence fastest. Empirical validation confirms 0.2 as our method's optimal reduction ratio.

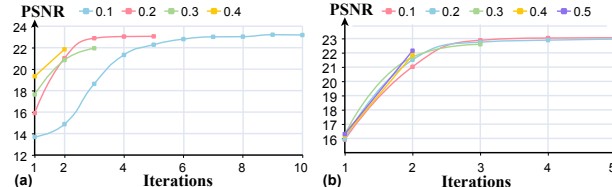

Figure 5: (a) Relationship between 3DGS rendering quality and noise reduction ratios in diffusion. (b) Relationship between 3DGS rendering quality and increasing ratio of $\vartheta$.

**Impacts of $\vartheta$ in equation 3.** We analyze mask prior constraint control via $\vartheta$ increase strategies (0.1-0.5), initialized at $\vartheta = 0$ with 0.1 per-iteration increments. Figure 5 (b) demonstrates iterative PSNR improvements across all ratios. While higher ratios enhance early-stage reconstruction, values bigger than 0.1 trigger premature inpainting confidence, causing insufficient VISTA-GI/CL module interaction and subsequent performance degradation.

**Comparisons of different methods in extreme cases.** To validate our advantages in the extreme case with large viewpoint differences (sparse views), we quantitatively evaluate various methods for the extreme case, and the results are as Table 4. Our method still outperforms existing methods in removing dynamic distractors under such extreme conditions.

| Method | LPIPS ↓ | PSNR ↑ | SSIM ↑ |
|---|---|---|---|
| InFusion | 0.23 | 19.34 | 0.78 |
| SPIn-NeRF | 0.15 | 23.33 | 0.82 |
| SpotLess | 0.14 | 24.75 | 0.84 |
| Ours | **0.10** | **26.38** | **0.86** |

Table 4: Comparison of different methods in the extreme case.

## 6    CONCLUSION

In this work, we presented VISTA, a novel framework for 3D Gaussian inpainting that leverages complementary visual and semantic cues from multiple input views. By introducing visibility uncertainty maps and integrating visibility-uncertainty-guided inpainting (VISTA-GI) with scene conceptual learning (VISTA-CL), our method addresses key challenges in both static and dynamic 3D scene editing. Experiments on the SPIn-NeRF and UTB180-derived datasets demonstrate that VISTA outperforms state-of-the-art methods, producing high-quality 3D reconstructions with seamlessly filled masked regions and effective distractor removal. Its ability to handle complex inpainting scenarios and dynamic objects makes it a versatile tool for augmented and virtual reality, advancing the pursuit of seamless, realistic 3D scene editing.

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

# A    APPENDIX

**Use of Large Language Models (LLMs)**    We acknowledge the use of a large language model as a general-purpose assistive tool. Its role was limited to language polishing and minor stylistic refinement of the manuscript. The LLM was not involved in research ideation, experimental design, data analysis, or substantive writing. All scientific content, including methodology, results, and conclusions, was conceived, implemented, and written entirely by the authors.

**Experiment Setup**    Our 3D reconstruction and 2D inpainting method is implemented on a single RTX 4090. We use the default parameters of 3DGS for reconstruction, generating a reconstructed render every 10,000 iterations. Additionally, we employed the commonly used Stable Diffusion v1.5 (Rombach et al., 2022) as the base inpainting model, training it for 3,000 iterations using textual inversion for scene representation. Our diffusion model inference consists of a 50-step denoising process, initialized with a noise strength of 1.0 that is progressively reduced by a factor of 0.2 at each iteration.

**Time cost analysis and comparison.**    To quantitatively evaluate performance and computational efficiency, we compare our method against baseline approaches (InFusion, SPIn-NeRF, and Spot-Less) on the synthetic scene shown in Figure 9. This scene provides ground truth data, enabling evaluation through reference-based metrics for both rendering quality and computational efficiency during optimization.

| Method | LPIPS ↓ | PSNR ↑ | Time Cost |
|---|---|---|---|
| InFusion | 0.23 | 19.34 | 16m 34s |
| SPIn-NeRF | 0.15 | 23.33 | 7h 32m 18s |
| SpotLess | 0.14 | 24.75 | 30m 26s |
| Ours | 0.10 | 26.38 | 33m 34s |

Table 5: Quantitative results and time costs on the synthesis data.

As shown in Table 5, while our method incurs additional computational overhead compared to vanilla 3DGS due to the integration of iterations and diffusion models, it achieves superior rendering quality while maintaining comparable efficiency to state-of-the-art 3DGS methods (e.g., SpotLess (Sabour et al., 2024)). Furthermore, our approach demonstrates significantly better reconstruction quality while being approximately 10× faster than leading NeRF-based methods such as SPIn-NeRF.

**Reasons for combining raw images $\mathcal{I}$ and $\tilde{\mathcal{I}}$ rather than substituting raw images $\mathcal{I}$ with $\tilde{\mathcal{I}}$ in Section 4.4**    As shown in Figure 6, the reconstruction without raw images could not render the seaweed without ambiguity. The accumulated error from two iterations, caused by 3DGS's inability to fit the scene fully and the uncertainty introduced by the generated model, deteriorates the image quality. Raw images act as an "anchor" for our method, ensuring that the rendered images align closely with the input images and do not deviate significantly.

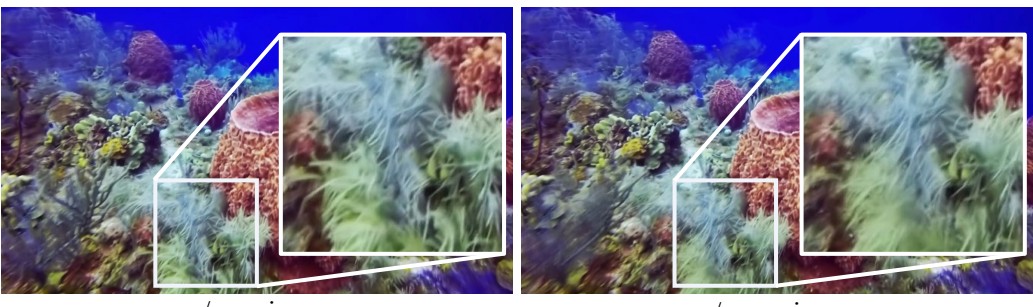

| w/ raw images | w/o raw images |

Figure 6: Reconstruction results with and without raw images. Involving the raw images in our method will improve the inpainting performance.

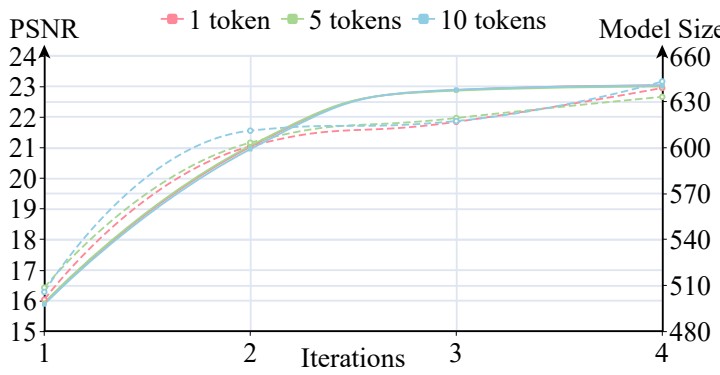

Figure 7: Relationship between model performance (PSNR) and model size (MB) with different token numbers. The dashed and solid lines represent the model size and performance variations, respectively. The model performance (solid lines) under different token numbers almost overlaps.

**Effects of token numbers to depict one scene.** As shown in Figure 7, our method balances scene depiction efficiency by optimizing token quantity (linked to color specificity) and textual inversion for coarse semantic learning, as excessive tokens inflate model size without performance gains. Training stabilizes at three iterations (determined by PSNR smoothing), beyond which redundant Gaussians overfit diffusion noise, further increasing model parameters.

## A.1 IMPACT OF PRIOR (MASK / VIEWS) ON INPAINTING

**Impact of the mask quality on Inpainting** . Adding the prior (mask) information in our method will significantly improve the inpainting results, especially for those static objects. This is easy to understand because dynamic objects create inconsistencies during the reconstruction process, which our algorithm can detect. In contrast, static object inpainting necessitates the semantic information that the detection model identifies.

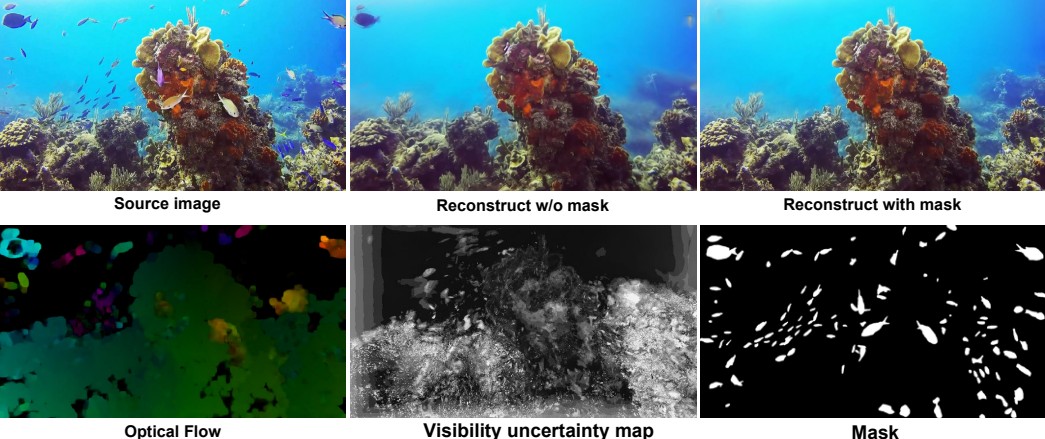

Figure 8: Impact of prior (mask) on inpainting results. Our method will improve inpainting performance by incorporating the mask information. To analyze the results, we also display the optical flow of the source image and the visibility uncertainty map.

For instance, in the top-left corner of Figure 8, the fish is retained while the others are removed. This is primarily because the fish remains stationary across different views (as evident in the optical flow map of Figure 8, where the top-left fish exhibits low flow values at its center). Consequently, it has a lower value in the visibility uncertainty map (see the corresponding map in Figure 8). Without using a mask to label this area for repair manually, the fish's geometric characteristics resemble those of a stationary object, such as a rock, making it indistinguishable from our uncertainty detection system.

| Mask Quality | LPIPS ↓ | PSNR ↑ | SSIM ↑ |
|---|---|---|---|
| 100 % | 0.10 | 26.38 | 0.86 |
| 50 % | 0.10 | 25.73 | 0.81 |
| 0 % | 0.12 | 25.06 | 0.79 |

Table 6: Quantitative results of large viewpoint differences.

In contrast, moving fish create significant geometric inconsistencies across viewpoints, enabling our uncertainty detection to flag them as anomalies. This leads to their removal through the inpainting process. To address these challenging scenarios, we introduced mask annotations for fish detection, providing semantic guidance for our inpainting method. As shown in the last column of Figure 8, incorporating the mask ensures the successful removal of the top-left fish.

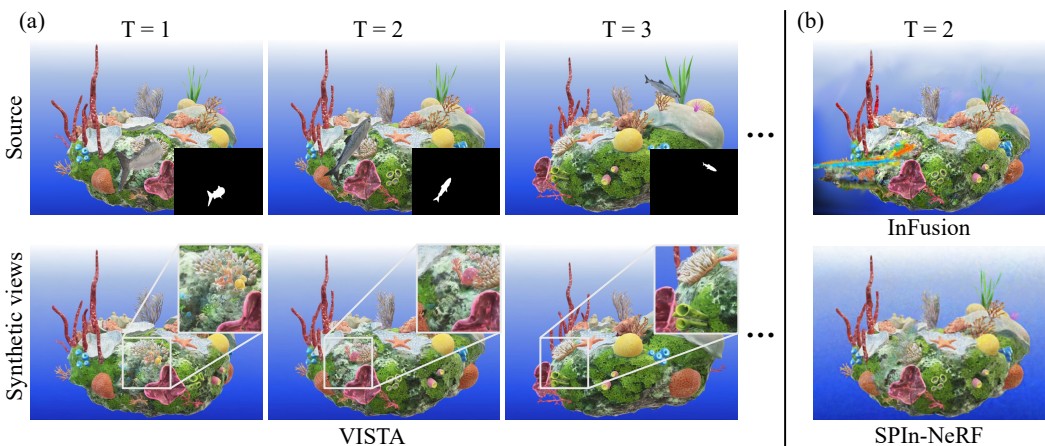

Figure 9: **(a) The figure of an artificial synthesis scene in extreme cases.** The original views of three adjacent cameras and the inpainting results of our method are demonstrated for comparison. **(b) The results of InFusion and SPIn-NeRF in extreme cases.** Their results are obtained by the camera from 'T = 2' in (a).

**The ablation study of mask quality.** Building upon the visualization of the missing mask prior presented earlier, we performed a rigorous quantitative analysis to validate the robustness of our proposed method. In particular, we conducted a controlled ablation study to systematically evaluate its performance under three distinct mask quality scenarios:

- **100% mask:** Complete masks from MASA detection as used in our method.
- **50% mask:** Randomly dropped 50% of the masked pixels from the complete masks.
- **0% mask:** No mask information provided.

We set three different mask qualities. Ablation studies (Table 6) show our method maintains stable performance when progressively removing object masks (100% → 50% → 0% mask). Metrics like PSNR dropped only 0.65 with 50% mask missing, demonstrating inherent robustness to mask absence. As shown in Figure 2 of the paper, this robustness is derived from iterative mask refinement utilizing the iterative refined uncertainty as an updated mask.

**Ablation study of large viewpoint differences** To evaluate the impact of viewpoint difference, we capture 34 images from continuously distributed viewpoints around a scene to create a ground truth (GT) 3DGS model. We systematically reduce the number of viewpoints by sampling them at different intervals $\{2, 3, 4, 5, 6, 7\}$, where larger intervals represent larger viewpoint differences. We reconstruct a 3DGS model per sampling interval and establish a quantitative analysis framework linking viewpoint differences to reconstruction quality by benchmarking rendered images

| Sampling Interval | LPIPS ↓ | PSNR ↑ | SSIM ↑ |
|---|---|---|---|
| 2 | 0.09 | 26.25 | 0.89 |
| 3 | 0.14 | 23.42 | 0.83 |
| 4 | 0.16 | 22.42 | 0.80 |
| 5 | 0.27 | 18.09 | 0.65 |
| 6 | 0.25 | 18.71 | 0.69 |
| 7 | 0.41 | 15.66 | 0.57 |

Table 7: Quantitative results of large viewpoint differences.

against GT models. Our method ensures robust reconstruction under significant viewpoint variations through view-consistency detection and repair mechanisms, while metrics degrade in extreme scenarios with missing key viewpoints due to irreversible information loss.

**VISTA in limited scenarios** Our uncertainty maps are built by observing a set of adjacent perspectives/views, thus fully utilizing complementary visual cues. However, in some extreme conditions (A sparse 360° capture using only eight input views evenly spaced around the object, yielding an average angular separation of 45°, which severely challenges reconstruction and editing), we don't have enough valid adjacent perspectives/views to get the visual cues. To investigate the performance of our method under such extreme conditions, we manually synthesized an extreme scenario where the camera rapidly changes poses, resulting in very few available adjacent viewpoints. In this case, the VISTA-GI can hardly detect the inconsistency between different views, requiring VISTA-CL to produce better results.

As shown in Figure 9 (a), thanks to the 2D diffusion model, our method utilizes its results to effectively inpaint the scene in such extreme conditions. Meanwhile, as shown in Figure 9 (b), the InFusion result is unrealistic due to neglecting consistency in inpainting. SPIn-NeRF shows a reasonable result, but with blurry and indistinguishable inpainting areas. Compared to other methods, our approach benefits from Scene Conceptual Learning, resulting in clearer and more reasonable repairs in the target areas, and the textures and content maintain consistency with the original scene.

## A.2 VISUALIZATION OF OUR METHOD

In this part, we visualize more results to demonstrate the effectiveness of our method and the potential failure scenarios that may arise.

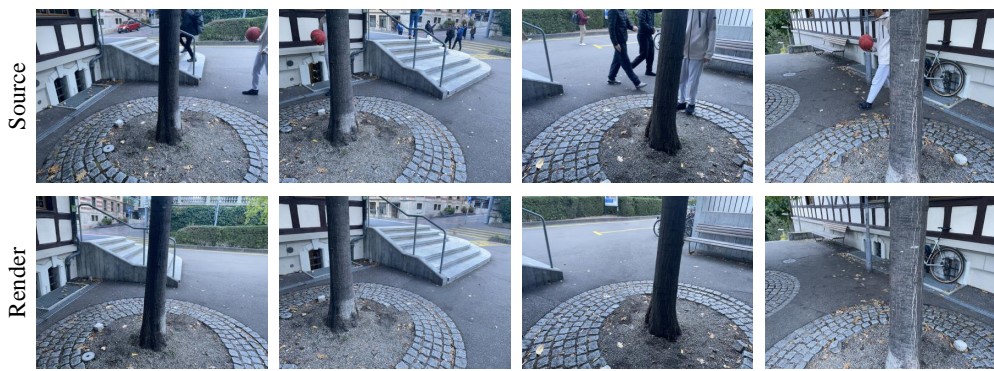

Figure 10: Case of real-world pedestrian removal from the nerf-on-the-go dataset.

**Real-world case.** The underwater dataset we used is derived from real-world diving videos, and due to the effects of the underwater medium and floating debris, these scenes are challenging scenarios in the real world. We also tested our dataset on a scene related to pedestrian removal from the NeRF-on-the-Go dataset. This scene, called Tree, contains 212 images, with the main distractors from moving pedestrians. As shown in Figure 10, our method achieved high-quality results on this

dataset. Due to the abundance of viewpoints in the dataset, there is a lot of supplementary information between perspectives, allowing our method to effectively utilize other viewpoints to repair the blurring caused by moving distractors.

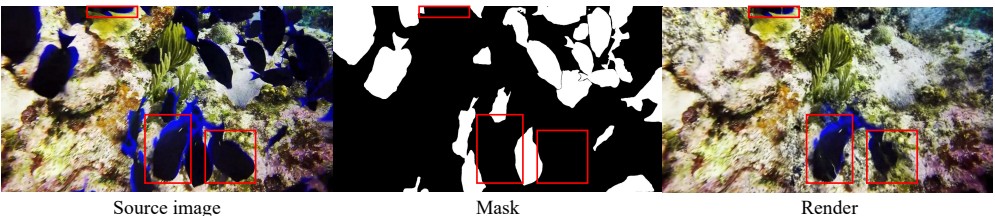

Source image           Mask           Render

Figure 11: Failure case from our underwater 3D inpainting dataset.

**Fail case from our dataset.** We provide failure cases of our algorithm in Figure 11. Due to errors in the prior mask, some fish were not detected by the object detection model. Furthermore, since the fish did not move significantly during the shooting process, these areas did not produce inconsistencies across multiple viewpoints during reconstruction, making it difficult for the VISTA-GI component to identify these areas through uncertainty. This also validates our algorithm design approach: VISTA-CL introduces semantic information through masks, while VISTA-GI incorporates geometric information through uncertainty, complementing each other to remove distractors. However, in this failed case, issues arose in both aspects, resulting in poor reconstruction quality of the final scene.

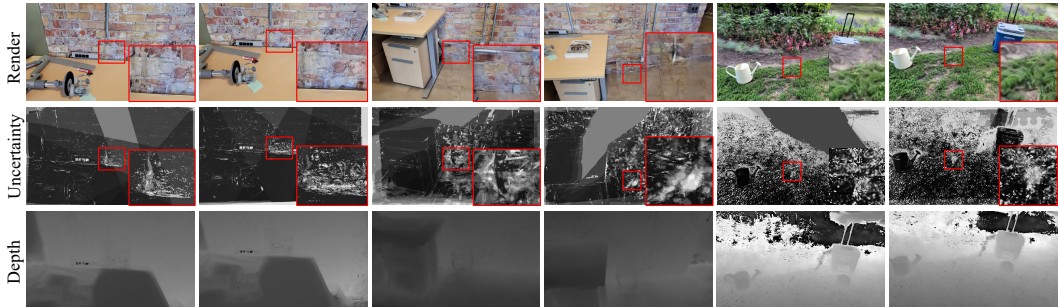

Figure 12: Visualization of the uncertainty map and depth of static scenes.

**Uncertainty and depth maps of static scenes.** As shown in Figure 12, we further visualize the uncertainty and depth maps of the static scenes. The deeper the color, the closer the depth. It can be observed that our method identifies areas in the rendered image that are inconsistent with other viewpoints and generates reasonable content.

| Resolution | LPIPS ↓ | PSNR ↑ | SSIM ↑ |
|---|---|---|---|
| 64×64 | 0.51 | 16.27 | 0.68 |
| 128×128 | 0.42 | 18.89 | 0.69 |
| 256×256 | 0.26 | 21.33 | 0.71 |
| 512×512 | 0.11 | 26.04 | 0.84 |
| 1299×974 | 0.10 | 26.38 | 0.86 |

Table 8: Quantitative ablation results of different resolutions.

## A.3    IMPACTS OF DIFFERENT IMAGE RESOLUTION

In our experiment setup, we use the stable diffusion v1.5 as the inpainting model and train and test the model following its default setup: if the input image has a resolution higher than 512×512, we

crop the image to a new size that is both the closest to the original image size and a multiple of 8; if the input image is smaller than 512×512, we rescale the image to 512×512. To analyze the influence of the strategy on different original resolutions, given an original scene with input images having a size of 1299×974, we downsample these images to four resolutions: 64×64, 128×128, 256×256, and 512×512. Then, for each resolution, we can build a 3D model and evaluate the rendering quality. As shown in Table 8, we observe that: (1) reducing the resolution to 512×512 does not significantly impact any of the metrics, demonstrating our method's robustness to substantial resolution changes. (2) further decreasing the resolution leads to gradual degradation in reference-based metrics, while non-reference metrics remain relatively stable.

