# OpenReview forum: "Visibility-Uncertainty-guided 3D Gaussian Inpainting via Scene Conceptional Learning"
_ICLR.cc/2026/Conference — ICLR 2026 Conference Withdrawn Submission_

### Official Review · Reviewer_KCnq · 2025-10-23

**Soundness:** 2
**Presentation:** 2
**Contribution:** 2
**Rating:** 4
**Confidence:** 4

**Summary:**

This paper proposes a novel framework VISTA for recovering missing regions in 3D scenes represented by Gaussian Splatting. The method introduces a VISTA-GI that models uncertain areas in occluded or incomplete views, guiding the inpainting process at both geometric and appearance levels. Furthermore, a VISTA-CL module integrates semantic priors to improve consistency and plausibility of reconstructed regions. The paper demonstrates improved completeness and rendering quality across 3D object inpainting and  dynamic distractor removal compared with existing 3DGS-based and NeRF-based inpainting methods.

**Strengths:**

（1）The paper introduces a visibility uncertainty modeling mechanism within 3D Gaussian Splatting, which is novel in the context of 3D scene completion.

（2）The paper is well-organized, and the writing is clear. The framework is clearly explained with helpful visualizations.

（3）The proposed framework is supported by extensive experiments across 3D object inpainting and dynamic distractor removal.

**Weaknesses:**

（1）Although the proposed method performs well in removing dynamic distractors, the results presented in Figure 4 indicate that the performance on large object removal still not good, with noticeable artifacts in inpainted regions.

（2）The experimental evaluation includes only a limited set of Gaussian-based inpainting baselines. To provide a more comprehensive comparison, the authors should also consider recent diffusion-based 3D inpainting approaches, such as GaussianEditor and TIP-Editor, which represent the current state of the art in 3D scene completion and editing.

（3）The proposed module appears overly heuristic. Overall the novelty lacks a clear theoretical basis for how visibility uncertainty is estimated and propagated throughout the pipeline.

（4）While the authors mention a comparison of overall time expenditures in the supplementary materials, the computational overhead of the VISTA-GI and VISTA-CL modules is not reported. A quantitative analysis of their runtime or memory cost would strengthen the paper’s completeness.

（5）The paper title is missing.

**Questions:**

（1）How is the visibility uncertainty quantitatively validated? Could you provide visualizations of uncertainty maps or a correlation study between uncertainty and reconstruction error?

（2）Can you compared your approach to more diffusion-based Gaussian inpainting methods?

（3）What pretrained semantic model is used in the Scene Conceptual Learning module, and how sensitive is the performance to its choice?

（4）What is the memory cost compared to standard and recent 3DGS inpainting frameworks?

---

### Official Review · Reviewer_Ye5c · 2025-10-31

**Soundness:** 2
**Presentation:** 2
**Contribution:** 2
**Rating:** 4
**Confidence:** 4

**Summary:**

This paper proposes VISTA, a framework for 3D Gaussian inpainting guided by visibility uncertainty and scene conceptual learning. VISTA consists of two core components: VISTA-GI and VISTA-CL. VISTA-GI introduces a method to compute the visibility uncertainty of 3D points by measuring color variations across different views, which is then used to guide the optimization of the 3DGS. VISTA-CL leverages a diffusion model to learn a semantic concept of the scene, enabling high-quality inpainting of regions lacking visual cues from other views. VISTA is evaluated on the SPIn-NeRF and an underwater dataset derived from UTB180.

**Strengths:**

1. The method's capability in handling dynamic objects for 3D inpainting is a notable strength.
2. The experiments demonstrate good performance.
3. The paper is well-structured and easy to follow.

**Weaknesses:**

1. While VISTA-GI's use of color variation to compute visibility uncertainty performs well in dynamic underwater scenes, its effectiveness in purely static scenes requires further discussion.
2. Since VISTA-CL relies on 2D diffusion-based inpainting, how is multi-view consistency ensured for the generated content? Furthermore, could this approach introduce more artifacts when handling large masked regions?
3. The evaluation is primarily conducted on only two types of datasets. It would be beneficial to test the method's robustness in more diverse scenarios, such as static scenes with large viewpoint variations (e.g., Mip-NeRF 360 dataset with 360-degree scenes) and other dynamic environments like driving scenes (the KITTI dataset).

**Questions:**

Please refer to the weaknesses.

---

### Official Review · Reviewer_AH9W · 2025-10-31

**Soundness:** 3
**Presentation:** 3
**Contribution:** 3
**Rating:** 4
**Confidence:** 4

**Summary:**

This paper proposes VISTA for 3D Gaussian inpainting by combining visibility uncertainty guided 3D optimization with concept learning driven diffusion inpainting  The method computes per view uncertainty to weight multi view cues for re optimization and learns a scene concept without target objects via textual inversion to guide masked diffusion updates  Results on SPIn NeRF static scenes and an underwater dynamic dataset show improvements over several prior 3DGS and NeRF based baselines.

**Strengths:**

- Clear decomposition into GI and CL modules with an iterative scheme
- Visibility uncertainty offers an intuitive way to exploit complementary cross view evidence
- Demonstrates both static object removal and dynamic distractor cleaning with ablations

**Weaknesses:**

- Lacks comparison with the latest 3D inpainting methods, such as AuraFusion360 and depth guided cross view consistency 3DGS inpainting, especially on 360 USID dataset
- Heavy reliance on external masks with limited robustness analysis under imperfect detection and segmentation
- Temporal metrics are weak  the evaluation only includes InFusion SPIn NeRF and SpotLess without reporting temporal stability comparisons on SPIn NeRF dataset or 360 USID dataset


[1] AuraFusion360: Augmented Unseen Region Alignment for Reference based 360° Unbounded Scene Inpainting, CVPR 2025

[2] 3D Gaussian Inpainting with Depth Guided Cross View Consistency, CVPR 2025

**Questions:**

1. Can the authors add results on 360 datasets including AuraFusion360 and depth guided 3DGS baselines
2. Please report temporal metrics such as stability and sequence level consistency on SPIn NeRF or 360 USID datasets
3. How robust is the method when external masks are noisy or incomplete

---

### Official Review · Reviewer_BrKT · 2025-11-02

**Soundness:** 2
**Presentation:** 3
**Contribution:** 3
**Rating:** 6
**Confidence:** 3

**Summary:**

The work proposed VISTA, a general 3DGS object removal framework for 3DGS scenes with static or dynamic transient objects to be removed. VISTA leverages visibility uncertainty maps derived from adjacent multi-view information to assist dynamic object and artifacts removal. In addition, scene concept can be learned from marked inter-views via textual inversion, and enhance inpainted images. Accordingly, VISTA progressively learns scene concepts from views' visibility and inpainted masked object in multi-views, preserving scene's identity.

**Strengths:**

1. VISTA utilizes complementary information from RGB masked inter-views to enhance 3DGS inpainting, through the proposed point-level visibility uncertainty as well as learning semantic concept of the scene without masked object via textual inversion technique.
2. VISTA supports not only static but also dynamic transient object removal in 3DGS inpainting, based on naive 3DGS reconstruction.
3. The work conducted 3DGI experiments for static object inpainting and dynamic distractors, proving VISTA working better on inpainting and preserving original scene content. The experiments of ablation study, hyperparameter analysis and extreme cases showcased VISTA's robustness.

**Weaknesses:**

1. There are some missed recent related works: 3D Gaussian Inpainting with Depth-Guided Cross-View Consistency (CVPR 2025); AuraFusion360: Augmented Unseen Region Alignment for Reference-based 360° Unbounded Scene Inpainting (CVPR 2025); In-N-Out: Lifting 2D Diffusion Prior for 3D Object Removal via Tuning-Free Latents Alignment (NeurIPS 2024).
Thus some descriptions with "lastest work/methods" should also change the wording.

2. For dynamic inpainting, the work only evaluated on underwater 3D inpainting dataset, unlike SpotLessSplats that experimented on diverse scenarios.

**Questions:**

1. Please add missed related works mentioned in Weaknesses in the main manuscript, and compare with them if applicable.
2. For dynamic inpainting evaluation and baseline comparison, it should experiments on more datasets such as those used in SpotLessSplats, for example the NeRF-on-the-Go dataset used in Appendix A.2.

Typo:
Line 291: "text-2-image" should be "text-to-image".
Table 2: "Fid" should be "FID"

---

### Note · Authors · 2025-11-23

I have read and agree with the venue's withdrawal policy on behalf of myself and my co-authors.